# Kinetic of the Antibody Response Following AddaVax-Adjuvanted Immunization with Recombinant Influenza Antigens

**DOI:** 10.3390/vaccines10081315

**Published:** 2022-08-14

**Authors:** Ted. M. Ross, Naveen Gokanapudi, Pan Ge, Hua Shi, Robert A. Richardson, Spencer R. Pierce, Pedro Sanchez, Subhan Ullah, Eliana De Luca, Giuseppe A. Sautto

**Affiliations:** 1Center for Vaccines and Immunology, University of Georgia, Athens, GA 30602, USA; 2Department of Infectious Diseases, University of Georgia, Athens, GA 30602, USA

**Keywords:** influenza virus, influenza vaccine, antibody, IgG subclasses, antibody-secreting cells (ASCs), adjuvant, AddaVax, computationally optimized broadly reactive antigens (COBRA), hemagglutinin (HA)

## Abstract

Notwithstanding the current SARS-CoV-2 pandemic, influenza virus infection still represents a global health concern in terms of hospitalizations and possible pandemic threats. The objective of next-generation influenza vaccines is not only to increase the breadth of response but also to improve the elicitation of an effective and robust immune response, especially in high-risk populations. To achieve this second objective, the administration of adjuvanted influenza vaccines has been considered. In this regard, the monitoring and characterization of the antibody response associated with the administration of adjuvanted vaccines has been evaluated in this study in order to shed light on the kinetic, magnitude and subclass usage of antibody secreting cells (ASCs) as well as of circulating antigen-specific serum antibodies. Specifically, we utilized the DBA/2J mouse model to assess the kinetic, magnitude and IgG subclass usage of the antibody response following an intramuscular (IM) or intraperitoneal (IP) immunization regimen with AddaVax-adjuvanted bivalent H1N1 and H3N2 computationally optimized broadly reactive antigen (COBRA) influenza recombinant hemagglutinins (rHAs). While the serological evaluation revealed a homogeneous kinetic of the antibody response, the detection of the ASCs through a FluoroSpot platform revealed a different magnitude, subclass usage and kinetic of the antigen-specific IgG secreting cells peaking at day 5 and day 9 following the IP and IM immunization, respectively.

## 1. Introduction

Influenza virus infection represents a global health concern due to the significant number of hospitalizations and associated deaths, especially among high-risks populations such as the elderly, children and immunocompromised individuals [1]. Vaccination represents the main countermeasure to prevent infection and spread by influenza virus and diminish the associated global health and economic burdens. The immune response elicited by the administration of the predominantly utilized current standard of care (SOC) influenza vaccines, the split inactivated (IIV, e.g., Fluzone) and the recombinant subunit (i.e., Flublok) influenza vaccines is primarily antibody-mediated and composed by immunoglobulins (Ig) derived from recalled memory B cells (MBC) [2]. In fact, the human population can be considered “universally” preimmune to influenza virus [3].

The antibody response following SOC influenza vaccination has been demonstrated to vary among different populations depending on their age and immunologic and metabolic status as well as their genetic background, with the elderly and immunocompromised subjects identified as those individuals showing a lower and narrower antibody response, and thus representing also the most vulnerable populations to influenza infection [4,5,6]. In addition to this limitation, the current SOC influenza vaccines are generally characterized by a variable and incomplete effectiveness, ranging between 40 and 60% among the overall population [7]. For the above reasons adjuvanted influenza vaccines have been considered for human administration, especially in high-risk populations.

An adjuvanted inactivated influenza vaccine (i.e., FLUAD) is currently approved for human use in different countries (including the US, Canada and 15 European countries) and recommended for subjects 65 years and older. This influenza vaccine contains MF59, an oil-in-water emulsion of squalene oil-based adjuvant [8].

The mouse model, and especially the DBA/2J mouse strain, represents an accessible and frequently used preclinical animal model to evaluate not only antiviral drugs for therapy but also to characterize the immune response to influenza infection or vaccination using seasonal, pandemic and next-generation influenza immunogens [9,10,11]. However, the tracking of the antibody response both at the serological and ASC levels following influenza vaccination and using different routes of immunizations has never been described so far using this animal model. This type of investigation is fundamental not only to understand the kinetic of the antibody response following influenza vaccination but also to provide investigators with a guidance on the proper time points for sample collection, for example for the generation of reagent tools (e.g., polyclonal and monoclonal antibodies) as well as for performing endpoint studies such as in the case of an infection challenge study.

AddaVax is a well-known and frequently used adjuvant for experimental immunizations including those involving influenza antigens (Ags) for vaccination purposes and for eliciting and evaluating the antibody response to influenza virus, especially in the context of influenza-naïve animals where multiple adjuvanted Ag administrations are needed to elicit a detectable antibody response as that seen in humans [9,12,13].

In this context, a plethora of monoclonal antibodies (mAbs) have been generated in mice to characterize and dissect the antibody response to influenza Ags as well as possible tools for developing prophylactic and therapeutic approaches for influenza infection [14]. In this regard, when generating polyclonal and monoclonal reagents it is important to identify the kinetic of the antibody response following an Ag exposure, both at the serological and at the antibody-secreting cell (ASC) level, in order to define the appropriate time points for the corresponding sample collection and their characterization. This is also important information to consider when performing challenge studies and collecting samples for performing antibody binding and functional studies. Additionally, the generation of B cell hybridomas represents the classical and commonly used methodology for generating mAbs from mice, and the success of generating such reagents strictly depends on multiple factors such as the cell viability and the magnitude of the antibody response, which also depends on identifying the right time point for the sample collection. In particular, the spleen represents the most easily accessible and B and T lymphocyte-enriched secondary lymphoid organ, and it is commonly used as a source of B cells for generating mouse B cell hybridomas.

The monitoring of antibody responses has usually largely relied upon serum immunoglobulin assessment due to the ease of sample collection and high antibody concentration in this compartment [15]. However, secreted antibody exhibits a relatively short half-life (~3 weeks) *in vivo*, and maintenance of serum antibody levels therefore requires continuous replenishment and stimulation [16,17]. Differently from after an acute Ag exposure, in which elevated frequencies of circulating short-lived plasma cells (SLPC) can be found and transiently alter the serum antibody reactivity [18,19], long-lived plasma cells (LLPCs) residing in the bone marrow and secondary lymphoid tissues are the main responsible under steady state for the composition and specificity of serum antibody [20]. LLPCs are thought to originate from terminally differentiated B cells that participated in T-cell-dependent germinal center (GC) reactions [21,22]. SLPC are also generated the same way; however, only a subset of these cells acquires the appropriate gene expression program and successfully occupies the specialized niche required for long-term survival and maintenance of sustained antibody production [23,24]. Such LLPCs and their secreted antibodies represent the first line of humoral defense against reinfection with pathogens such as seasonal influenza viruses and many others [25,26,27,28].

In this work, we utilized and tracked the polyclonal serum and spleen ASC kinetic, magnitude and IgG isotype distribution after the vaccination of mice with a mixture of previously described next-generation computationally optimized broadly reactive antigen (COBRA) influenza recombinant hemagglutinins (rHAs) [12,29] adjuvanted with AddaVax and administered through the two most commonly utilized routes in a preclinical setting (intraperitoneally (IP) and intramuscularly (IM)). In particular, these antigens were utilized as representative rHA for shedding light on the antibody response elicited by next-generation influenza vaccines in a preclinical setting, especially in view of their deployment in upcoming clinical trials.

## 2. Materials and Methods

### 2.1. Vaccination of Mice

DBA/2J mice (females, 6 to 8 weeks old) were purchased from the Jackson Laboratory (Bar Harbor, ME). The mice were housed in microisolator units and fed *ad libitum*. Mice were randomly divided into 16 groups (6 animals/group) and vaccinated IM or IP with a combination of 3 μg/Ag of COBRA Y2 (H1N1) and COBRA NG2 (H3N2) rHAs [12], expressed and purified as previously described [30]. All vaccinations, with the exception of the final boost, were formulated with an emulsified squalene-based oil-in-water emulsion adjuvant, AddaVax (InvivoGen, San Diego, CA, USA), and the final concentration after mixing 1:1 with rHA was 2.5% squalene in a total volume of 50 μL. The final boost was composed by the same quantity of COBRA rHAs resuspended in phosphate-buffered saline (PBS, Corning, Tewksbury, MA, USA) only. Vaccines were administered into the hind leg of the animals (for the IM immunization) or in the peritoneal cavity (for the IP immunization) on days 0, 21 and 42 in a homologous prime-boost-boost regimen. Blood was terminally collected from the facial vein at the day of the sacrifice after the final boost, specifically at day 3, 5, 6, 7, 8, 9, 12 and 14 after the final boost (Figure 1). Serum was isolated from the blood by centrifugation at 2500 rpm for 10 min. Clarified serum was removed and frozen at −20 ± 5 °C until usage. Spleens were also collected at the day of the sacrifice after the final boost (at day 3, 5, 6, 7, 8, 9, 12 and 14) and homogenized using a gentleMACS™ dissociator (Miltenyi Biotec, Bergisch Gladbach, Germany). Homogenized spleens were then washed with B-cell medium (BCM) composed by: RPMI 1640, 10% fetal bovine serum (FBS), 2 mM L-glutamine, 100 U/mL penicillin, 100 μg/mL streptomycin, 8 mM HEPES, 23.8 mM sodium bicarbonate, essential and nonessential amino acids, 1 mM sodium pyruvate and 50 μM 2-mercaptoethanol. All components of the BCM were purchased from Thermo Fisher Scientific (Waltham, MA, USA) with the exception of the FBS that was purchased from Atlanta Biologicals, Flowery Branch, GA, USA. Cells were then pelleted at 400× *g* for 7 min, resuspended in a solution of 90% FBS and 10% dimethyl sulfoxide (DMSO, Sigma-Aldrich, Darmstadt, Germany) and stored frozen in liquid nitrogen until usage. For the FluoroSpot assay (Section 2.4), splenocytes were thawed at 37 °C in a water bath, washed and resuspended in BCM and counted using a LUNA-II automated cell counter (Aligned Genetics, Inc., Anyang, South Korea). A splenocyte viability of ~90% was normally achieved following the above procedure.

### 2.2. ELISA

ELISA was used to assess the level of serum antibody reactivity against the corresponding rHA Ags utilized for the immunization. COBRA Y2 and NG2 rHAs were used for all the binding experiments. ELISA was performed as previously described [13]. In brief, Immulon 4HBX plates (Thermo Fisher Scientific) were coated overnight at 4 °C with 50 μL per well of a PBS solution (pH 9.4) containing 1 μg/mL COBRA Y2 or NG2 rHA in a humidified chamber. Plates were blocked with 200 μL per well of blocking buffer for at least 1 h at 37 °C. The mouse sera were 3-fold serially diluted in blocking buffer starting from 1:100, and plates were incubated for 1 h at 37 °C. Plates were washed five times with PBS; 100 μL per well of HRP-conjugated goat anti-mouse IgG1, or IgG2a or IgG2b (Southern Biotech, Birmingham, AL, USA) diluted in blocking buffer at 1:4,000, 1:2,000 and 1:500, respectively, were added; and plates were incubated for an additional 1 h at 37 °C. Finally, plates were washed five times with PBS, and 2,2′-azino-bis(3-ethylbenzothiazoline-6-sulfonic acid) (ABTS) substrate (VWR International, Radnor, PA, USA) was added, and plates were incubated at 37 °C for 15–20 min. Colorimetric conversion was terminated by addition of 1% SDS (50 μL per well), and OD was measured at 414 nm (OD414) using a spectrophotometer (PowerWave XS; BioTek, Winooski, VT, USA). For IgG, subclass quantification-obtained OD414 values were interpolated through a nonlinear regression analysis with those obtained on IgG1-, IgG2a- and IgG2b-specific standards (BioLegend, San Diego, CA, USA).

### 2.3. Hemagglutination Inhibition Assay

The hemagglutination inhibition (HAI) assay was used to assess functional antibodies to the HA able to inhibit hemagglutination of turkey (for H1N1) or guinea pig (for H3N2) erythrocytes. HAI assay was performed as previously described [6,13]. In detail, to inactivate nonspecific inhibitors, sera were treated with receptor-destroying enzyme (RDE) (Denka Seiken, Tokyo, Japan) prior to being tested. Briefly, three parts of RDE were added to one part of sera and incubated overnight at 37°C. RDE was inactivated by incubation at 56 °C for 30 min. RDE-treated sera were diluted in duplicate in a series of 2-fold serial dilutions in v-bottom microtiter plates (Greiner Bio-One, Kremsmünster, Austria). An equal volume of the H1N1 (A/California/07/2009) or H3N2 (A/Texas/50/2012) virus adjusted to ∼8 hemagglutination units/50 μL was added to each well. As per protocols of the WHO laboratory influenza surveillance manual, the H3N2 virus dilution was prepared in the presence of 20 nM oseltamivir carboxylate. The plates were covered and incubated at room temperature for 20 min, and then 0.8% of turkey or 0.75% of guinea pig erythrocytes (Lampire Biologicals, Pipersville, PA, USA) in PBS were added and incubated at room temperature for 30 min or 1 h, respectively. Erythrocytes were stored at 4 °C and used within 72 h of preparation. The HAI titer was determined by the reciprocal dilution of the last well that contained nonagglutinated erythrocytes. Positive and negative serum controls were included for each plate. All mice were negative (HAI ≤ 1:10) for preexisting antibodies to currently circulating human influenza viruses prior to vaccination.

### 2.4. FluoroSpot Assay

For enumeration of spleen ASCs, irrespective of their Ag specificity, splenocytes cell suspensions were serially diluted 2-fold in quadruplicates, starting at 1 × 10^5^ live cells/well, in round-bottom 96-well tissue culture plates (Corning, Corning, NY, USA) and subsequently transferred into 70% (*v*/*v*) ethanol preconditioned assay plates coated with anti-Igκ/λ capture antibody contained in the mouse IgM/IgG1/IgG2a/IgG2b Four-Color ImmunoSpot^®^ kit (CTL, Shaker Heights, OH, USA), following the instructions provided by the manufacturer. Plates were incubated for 16 h at 37 °C, 5% CO_2_, and plate-bound Ig spot-forming units (SFU), each representing a secretory footprint of an individual ASC, were revealed using the IgM-, IgG1-, IgG2a- and IgG2b-specific detection reagents contained in the kit, which was used according to the manufacturer’s instructions. For enumeration of Ag-specific ASCs, COBRA Y2 or COBRA NG2 rHA proteins were coated directly into 70% (*v*/*v*) ethanol preconditioned assay wells at 25 μg/mL in PBS overnight at 4 °C in a humidified chamber. Following one wash with 150 μL PBS, plates were blocked with 150 μL of BCM for 1 h at room temperature. Mouse splenocytes were then added starting from 3 × 10^5^ cells/well in quadruplicates and 2-fold serially diluted. Plates were then incubated for 16 h at 37 °C, 5% CO_2_, and SFU were visualized using the mouse IgM/IgG1/IgG2a/IgG2b Four-Color ImmunoSpot^®^ kit (from CTL) according to the manufacturer’s instructions. Following completion of B cell ImmunoSpot^®^ assay detection systems, plates were air-dried prior to scanning on an ImmunoSpot^®^ S6 Ultimate Analyzer. SFU were then enumerated using the Basic Count mode of CTL ImmunoSpot SC Studio (Version 1.6.2, Shaker Heights, OH, USA).

### 2.5. Statistical Analysis

Comparisons of the binding (ELISA) and functional (HAI) serum antibody titers as well as of the enumerations of the ASCs at the different time points following the final boost using the IM and IP immunization routes were investigated by two-way ANOVA using GraphPad Prism 9.3.1 (RRID: SCR_002798, San Diego, CA, USA). Obtained *p* values equaling or lower than 0.05 were considered statistically significant. Synopsis of all the obtained *p* values are reported in the Appendix A.

## 3. Results

### 3.1. Predominant Serum IgG1 Response following IM or IP AddaVax-Adjuvanted Vaccination

Binding assays performed in ELISA using sera collected following the final boost of mice revealed that the COBRA Y2 and NG2 Ag-specific polyclonal antibody responses are mainly composed by IgG1 antibodies at all the tested time points (3–14 days following the final boost). This predominant IgG1 antibody response is observed both in IM- and IP-vaccinated mice with an average 1–7.5 and 2.5–10 mg/mL of serum IgG1 equivalents against COBRA Y2, and of 1–4 and 1.5–6 mg/mL against COBRA NG2, respectively (Figure 2). Additionally, this IgG1 antibody response follows a stable and constant trend, characterized by some minor but statistically significant (*p* < 0.05) Ag-specific IgG1 peaks across all the time points (Figure 2 and Appendix A). On the other end the COBRA Y2- and NG2-specific IgG2a and IgG2b antibody response is of lower magnitude both in IM- and IP-vaccinated mice with an average 20–100 and 20–175 μg/mL of serum IgG2a and IgG2b equivalents against COBRA Y2, and of 10–150 and 10–100 μg/mL against COBRA NG2, respectively. Differently from the IgG1 serum antibody response, the Ag-specific IgG2a and IgG2b is of lower magnitude, and the observed variability across the different time points could be attributed to the internal assay variability, considering the absence of a statistically significant difference (*p* > 0.05) of the IgG2a and IgG2b values across all the time points. A synopsis of the statistical analysis comparing the IgG1, IgG2a and IgG2b antibody subclass titers across the different time points is reported in the Appendix A.

### 3.2. Serum Hemagglutination Inhibition Activity Is Not Affected by the Time Point of Sample Collection

The same serum samples utilized for evaluating the antibody titer in terms of Ag binding were also used to assess their functional titer in terms of HAI activity against representative H1N1 and H3N2 virus strains for COBRA Y2 and NG2, respectively. The breadth of HAI activity elicited by AddaVax-adjuvanted COBRA Y2 and NG2 against historical and recent H1N1 and H3N2 vaccine strains has been previously reported [12]. As depicted in Figure 3, sera collected at the different time points following the final boost had an average HAI titer ranging from 1:120 to 1:1,000 in the IM-vaccinated groups and of 1:500 to 1:1000 in the IP-vaccinated groups against the H1N1 CA/09 strain. On the other hand, the HAI activity of both IM- and IP-vaccinated mice against the H3N2 TX/12 strain was of lower magnitude compared to CA/09. In particular, the IM-vaccinated mice showed an average HAI titer ranging from 1:10 to 1:20 while the IP-vaccinated groups showed an average HAI titer ranging from 1:30 to 1:130. No statistically significant differences were observed across the HAI titers against both the CA/09 and TX/12 strains using the serum samples collected at different time points (*p* > 0.05) within the analyzed day 3–14-time frame.

### 3.3. The Peak of Spleen ASCs Is Affected by the Route of Immunization

Splenocytes collected following the final boost were utilized for determining the kinetic, magnitude and IgG subclass distribution in the different mouse groups. As shown in Figure 4A,B the total antibody response was mainly composed by IgM, followed by IgG1, IgG2a and IgG2b ASCs in both the IM- and IP-immunized animal groups. The total number of IgM ASCs was higher across all the time points following the final boost, with an oscillatory trend, with a lower number of IgM ASCs at day 3 post the final boost in the IP-immunized group. Additionally, this oscillatory trend is characterized by some statistically significant (*p* < 0.05) IgM ASC peaks across some of the time points (Figure 4A,B and Appendix A). Total IgG1, IgG2a and IgG2b ASCs were overall lower compared to IgM ASCs and showing a higher number at days 3 and 9 in the IM-immunized group. This higher number of ASCs was more prominent for IgG1 by reaching a statistically significant (*p* < 0.05) difference when compared with the majority of the other time points (Figure 4A,B and Appendix A). A higher number of total IgG1 ASCs was also observed at day 5 in the IP-immunized group but without reaching a statistically significance (*p* > 0.05). When analyzing the Ag-specific (COBRA Y2 and NG2) antibody response, this was mainly composed by IgG1, IgG2a and IgG2b in the IM-immunized animals and predominantly by IgG1 ASCs in the IP-immunized animals (Figure 4C–F). This predominant Ag-specific IgG1 ASC response can be also appreciated by observing Figure 5. In particular, the IgG1 and IgG2a COBRA Y2- and NG2-specific ASCs peaked at day 9 (*p* < 0.05 for IgG1 when compared to all the other time points) following the final boost in the IM-immunized group (Figure 4C,E and Appendix A). Additionally, IgG2b ASCs showed a higher number at days 3 and 12 in the same group for COBRA Y2 (Figure 4C) that was statistically significant (*p* < 0.05) only when compared to day 7 IgG2b ASCs. Conversely, the IP-immunized group showed a statistically significant peak of IgG1 ASCs at day 5 (*p* < 0.05) with a lower extent of IgG2a and IgG2b ASCs across all the time points (Figure 4D,F). For both the IM- and IP-immunized groups, the extent of the ASC response against an irrelevant Ag (bovine serum albumin, BSA) was almost undetectable across all the time points (Figure 5) with the exception of the day 6 IgG2b ASCs in the IM-immunized group which averaged ~150 spots per million of input cells against all the 3 tested Ags (Y2, NG2 and BSA) and without reaching a statistical significance (*p* > 0.05). A synopsis of the statistical analysis comparing the magnitude of the IgM and IgG1, IgG2a, IgG2b antibody subclass ASCs across the different time points is reported in the Appendix A.

## 4. Discussion

In the context of influenza vaccination, adjuvants have shown to ameliorate immunity in patients typically having an impaired immune response upon vaccination, such as the infants and the elderly [31]. Most studies in the literature have been reported using alum as adjuvant, and it remains unknown how other adjuvants, such as squalene emulsion-based adjuvants (e.g., AF03, AddaVax, AS03 and MF59), act on germinal centers (GCs) and Ag release. This class of adjuvants has been demonstrated to elicit both cellular (Th1) and humoral (Th2) immune responses and to act through the recruitment and activation of antigen-presenting cells (APCs) and stimulation of cytokines and chemokines production by the innate immune system, specifically macrophages and granulocytes [31,32,33]. Additionally, they have been thought to protect the antigen, through the process of emulsification, from being bound and cleared by pre-existing serum antibodies which may allow for the draining of unprocessed antigen to lymph nodes [33,34]. By this means, influenza virus vaccines that use oil-in-water adjuvants are noted for their ability to activate and promote the affinity maturation of both naïve B cells and MBC [35,36]. Furthermore, a phase I clinical trial using a candidate next-generation influenza vaccine revealed that only subjects that received the AS03-adjuvanted IIV substantially seroconverted against the conserved HA stem domain [37]. Taken together, these studies indicate the critical dual role of adjuvants in promoting both a sustainable and broad immune response by not only boosting but also by promoting it against broadly reactive B cell epitopes of the HA, such as those that are part of the conserved stem region. In this regard, a study showed that the adjuvant AS03 induced an increased activation of naïve B cells and an increased adaptability of recalled MBC, improving immunogenicity [38]. Their recruitment and activation are typically triggered by an inflammatory response, which is the basis of most adjuvant strategies.

Similar and different classes of adjuvants have been or are currently evaluated in preclinical and clinical trials in combination with subunit and IIV vaccines, with the primary goals of eliciting a more robust immune response, increasing its breadth of activity and ultimately its breadth of protection as well as offering an Ag dose-sparing strategy, not only for high-risk individuals but also for the general population. As an example, AF03, a squalene-based emulsion adjuvant; Advax™, a polysaccharide adjuvant derived from delta inulin; and MAS-1, a water-in-oil emulsion-based adjuvant/delivery system comprised of stable nanoglobular aqueous droplets have been tested in preclinical and clinical studies and showed promising results [39,40,41].

In this context, considering their current utilization in humans (i.e., of MF59) for influenza vaccination, it is important to characterize the antibody response following vaccination with this class of adjuvants in order to shed light on the immunoglobulin magnitude, IgG subclass usage and kinetic both at the serological and ASC levels. In fact, this information is important to increase and improve our knowledge for establishing more accurate influenza animal models of immunity and help in the generation of tool reagents for influenza research.

The DBA/2J mouse strain is extensively used in the influenza field as a model for the influenza virus infection mainly because of its higher susceptibility as well as to assess the preclinical efficacy and validation of next-generation influenza vaccines and antibody response and, more recently, also in the context of vaccination for SARS-CoV-2 [10,42].

In this report, we thus utilized this mouse strain to evaluate the kinetic, magnitude and IgG subclass usage of the antibody response following the IM and IP administration of a combination of H1N1 and H3N2 next-generation influenza vaccines, namely COBRA Y2 and NG2, which have been previously demonstrated to elicit a broad antibody response both in influenza naïve and preimmune animal models, such as mice, ferrets and nonhuman primates (NHP) [13,43,44,45].

Consistently with previous reported observations, the AddaVax adjuvant, similar to other adjuvants belonging to the same class, elicits a predominant IgG1 antibody response in mice, especially when administered through the IP route, as shown in Figure 4 and as it can be visualized in Figure 5 [13]. Additionally, the DBA/2J mouse strain is more biased toward a Th2 response and thus a predominant IgG1 response [46]. Conversely, this IgG1 response is accompanied by both an IgG2a and IgG2b response when the vaccination is administered through the IM route. This phenomenon is consistent with previous observations and supported by the fact that the IM immunization is more effective in facilitating the influx of innate immunity components, such as monocytes that differentiate into APCs and polymorphonuclear neutrophils (PMNs) [47], which ultimately promote a more diverse IgG subclass distribution [34]. As previously demonstrated, this more diverse IgG subclass usage may be beneficial in terms of antibody Fc-mediated effector functions and consequently associated with a higher protection, since IgG2a, for example, is endowed with the highest affinity for the activating Fcγ receptor in mice. Additionally, the muscle remains the site of choice for most vaccinations due to its relative mass, ability to accommodate relatively large volumes of vaccine formulation and the ease of myocyte transduction (e.g., in the case of DNA or mRNA-based vaccinations) and transcription (e.g., MF59) [32]. As previously observed by our and other groups, the serological antibody response against H1N1 HAs is of higher magnitude compared to H3N2 HAs, either in terms of binding or functional activity, indicating an immunodominance of H1N1 HA compared to H3N2 HA, also when co-administered [48]. This difference may be attributed to the presence of more T_reg_ epitopes, especially in group 2 HAs as compared to group 1 HAs as well as to the structural properties (including glycosylation sites) of the two groups of proteins, which ultimately affect the serum antibody titer (both in terms of binding and functional activity) against the two Ag subtypes [49,50]. Conversely, the magnitude of Ag-specific ASCs in the spleen is comparable against the two proteins, and it follows a more stable trend along the different tested time points. This different trend and kinetic of the serum antibody response compared to that observed when analyzing the ASC component may be attributed to the longer half-life and persistence of the immunoglobulins that accumulated in the serum compartment [17,38].

It is noteworthy that the two vaccination routes (IM and IP) show a different peak of Ag specific IgG ASCs with the IM showing a delayed peak (at day 9) compared to the IP route (at day 5) and similarly to the time-point of the plasmablast peak observed following vaccination in humans [6,18,51,52]. This difference may be attributed to the different Ag release kinetic upon administration, with the IM route being associated with a slower Ag release as compared to the IP route. In fact, as already described, Ag is captured by the lymphatic system and transferred to the draining lymph nodes, the thoracic duct and hence the vascular system more rapidly following the IP as compared to the IM immunization [53,54]. Additionally, this slower release and persistence of the Ag may be also associated with a higher magnitude of Ag-specific ASCs (10-fold difference) in the IM-immunized group compared to the IP-immunized group, as previously described in other studies employing this route and the same class of adjuvants [55]. However, this difference is not reflected in a higher serum antibody titer since the two groups showed similar IgG1, IgG2a and IgG2b antibody levels.

One limitation of our work is that we did not assess the longevity of the MBC response, for example, upon further recalls happening farther in time from the initial vaccination regimen. Future studies will be aimed at better defining the MBC longevity and duration of the antibody response following vaccination with COBRA Ags as well as to compare other adjuvants and how they can improve the magnitude, breadth, longevity and duration of the antibody response. Another limitation is that, for easier accessibility and cellularity reasons, we assessed the presence of ASCs in the spleen only while we did not track them in other secondary lymphoid tissues such as the lymph nodes and in the bone marrow. Future studies will be aimed at investigating the kinetic and breadth of recognition of ASCs in these sites, since they represent the other main tissues where resident and LLPC reside, respectively. In addition, we also aim at performing these studies in the context of influenza virus infection as well as following vaccination in a preimmune setting. Finally, it will be important to perform similar studies using other classes of adjuvants and following their administration using also other type of routes (such as the subcutaneous and the intranasal routes) in order to better understand the associated immune mechanisms and kinetic of the antibody as well as of the T-cell-mediated immune responses. In this context, it will be important to evaluate also the IgA response at the serological and ASC level since the elicitation of this antibody class is mainly associated with the infection, such as in the case of influenza viruses, as well as with the vaccination through the mucosal route [13].

## 5. Conclusions

To conclude, in this work we described the kinetic, magnitude and IgG isotype usage of the antibody response at the serological and at the ASC level following an AddaVax-adjuvanted recombinant HA vaccination regimen utilizing a preclinical DBA/2J mouse model. We hope this information will be relevant for those groups investigating the antibody response following vaccination and provide indications for the determination of sample collection for studies aimed at evaluating and characterizing the B cell response.

## Figures and Tables

**Figure 1 vaccines-10-01315-f001:**
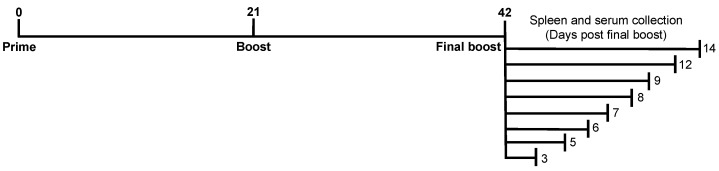
Schematic representation of the immunization regimen of the DBA/2J mice with the combination of COBRA Y2 (H1N1) and COBRA NG2 (H3N2) rHAs. AddaVax-adjuvanted Ags were administered intramuscularly IM or IP at day 0 and 21. Serum and spleens were harvested at the day of the sacrifice, specifically 3, 5, 6, 7, 8, 9, 12 and 14 days following the final boost with unadjuvanted Ags (at day 42) and collected serum and splenocyte samples utilized for ELISA or HAI and FluoroSpot assays, respectively.

**Figure 2 vaccines-10-01315-f002:**
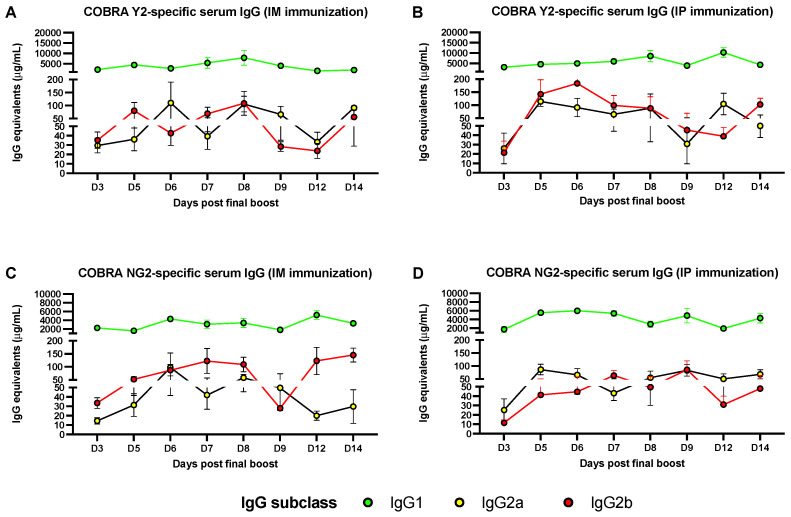
Quantification of Ag-specific IgG1, IgG2a and IgG2b from sera collected at different time points following the final boost of mice that were IM- (**A**,**C**) or IP-immunized (**B**,**D**) with the combination of COBRA Y2 and NG2 rHAs. The levels of the different Ag-specific immunoglobulins (IgG1, IgG2a and IgG2b) are expressed as IgG equivalents using a corresponding IgG subclass (IgG1, IgG2a and IgG2b) standard curve. Shown results are expressed as absolute mean values plus standard error of the mean (SEM) (error bars) using the different mouse group sera. The limit of detection of the assay is 0.5 ng/mL of IgG equivalents for each subclass. A synopsis of the statistical analysis comparing the IgG1, IgG2a and IgG2b antibody subclass titers across the different time points is reported in the Appendix A.

**Figure 3 vaccines-10-01315-f003:**
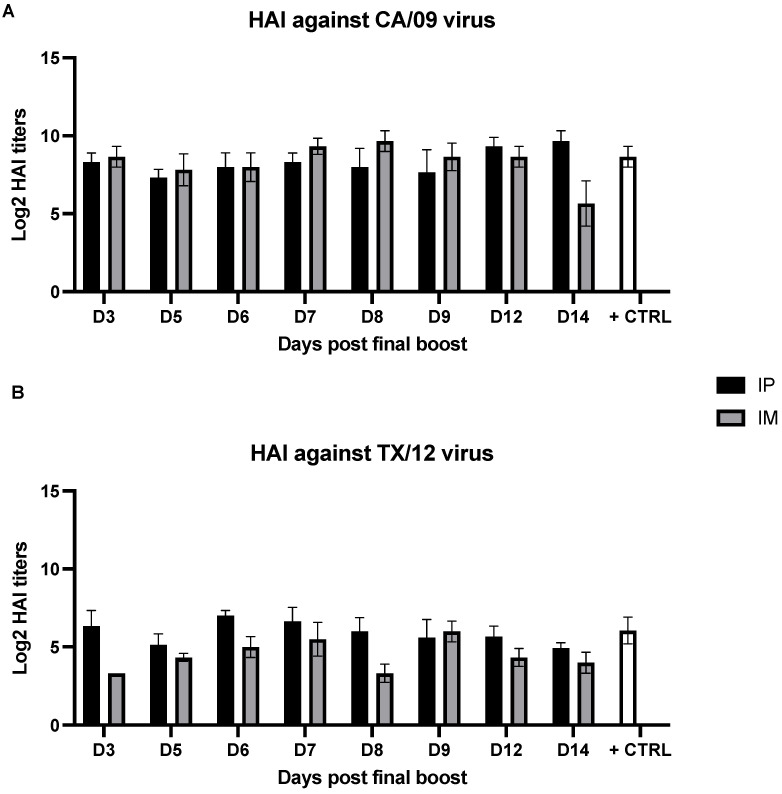
HAI from sera collected at different time points following the final boost of mice that were IP- or IM-immunized with the combination of COBRA Y2 and NG2 rHAs. The H1N1 A/California/07/2009 (CA/09) (**A**) and the H3N2 A/Texas/50/2012 (TX/12) (**B**) were used as representative virus strains to detect the HAI activity following the immunization with the COBRA Y2 and NG2 rHAs. Shown results are expressed as absolute mean values plus SEM (error bars) using the different mouse group sera. The limit of detection of the assay corresponds to a serum dilution of 1:10. Absence of statistically significant differences (*p* > 0.05) among the HAI titers of the sera collected at different time points against the CA/09 and TX/12 strains within the analyzed day 3–14-time frame was determined as specified in the Materials and Methods section.

**Figure 4 vaccines-10-01315-f004:**
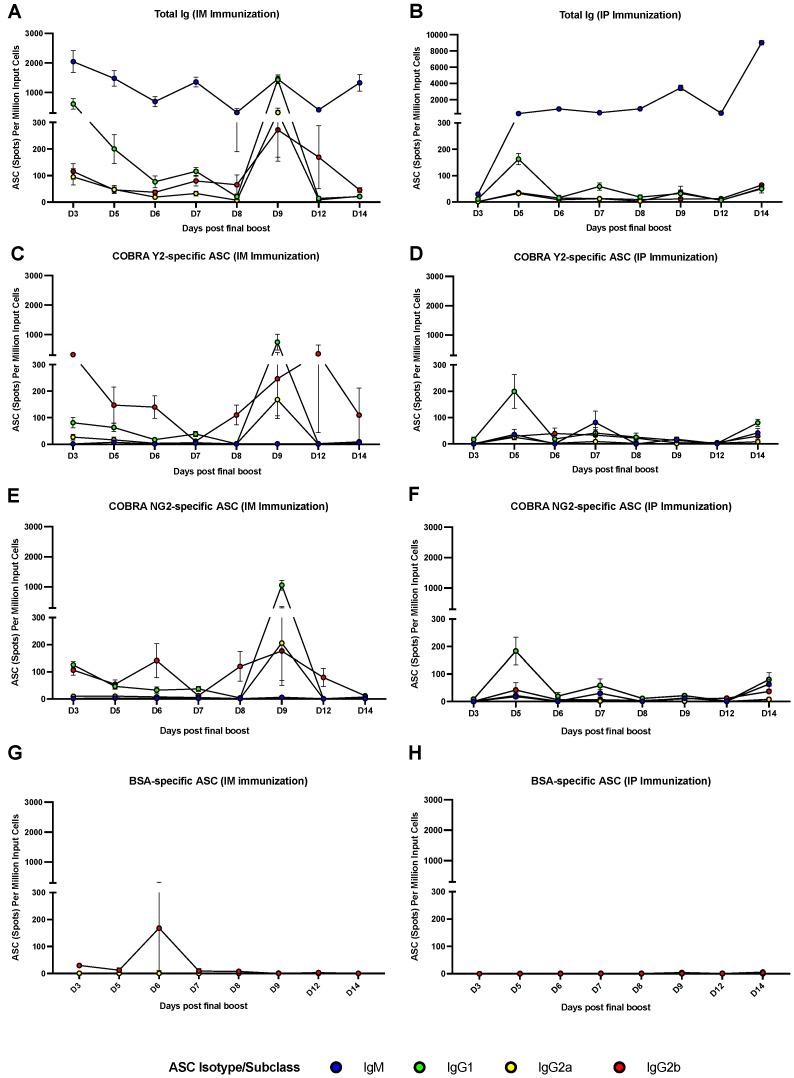
Enumeration of total (**A**,**B**), COBRA Y2 and NG2 Ag-specific ASCs (**C**–**F**) from splenocytes collected at different time points following the final boost of mice that were IP- or IM-immunized with the combination of COBRA Y2 and NG2 rHAs. BSA was used as an irrelevant antigen for ASC background reactivity purposes (**G**,**H**). Shown results are expressed as absolute mean values plus SEM (error bars) using the different mouse group splenocytes. The limit of detection of the assay is 1/100,000 cells (spots). A synopsis of the statistical analysis comparing the magnitude of the IgM and IgG1, IgG2a, IgG2b antibody subclass ASCs across the different time points is reported in the Appendix A.

**Figure 5 vaccines-10-01315-f005:**
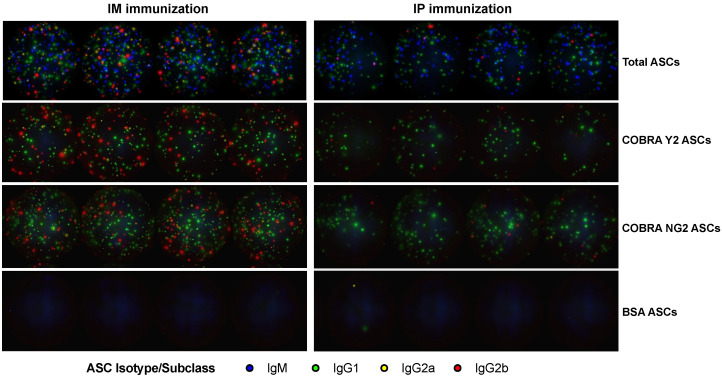
Representative FluoroSpot wells acquired from a 96-well plate for total (1.25 × 10^4^ splenocytes input/well), COBRA Y2 (7.5 × 10^4^ splenocytes input/well), COBRA NG2 (7.5 × 10^4^ splenocytes input/well) and BSA (3 × 10^5^ splenocytes input/well) ASCs from splenocytes collected following the final vaccine boost of mice through IM (left panels) or the IP (right panels) route using the combination of COBRA Y2 and NG2 rHAs. Represented wells were obtained from splenocytes collected at day 9 (for the IM-immunized group) and at day 5 (for the IP-immunized group) from one representative animal and run as quadruplicates.

## Data Availability

All data are included in the manuscript or can be obtained by contacting the corresponding author.

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
