# Peer review of "Kinetic of the Antibody Response Following AddaVax-Adjuvanted Immunization with Recombinant Influenza Antigens"

_vaccines, 2022, doi:10.3390/vaccines10081315_

Round 1

Reviewer 1 Report

A manuscript by T. Ross and co-authors presents the results of studying the kinetic, magnitude and IgG isotype usage of the antibody response in DBA/2J mice immunized with COBRA antigens adjuvanted with AddaVax. The authors measured the antibody responses at the serological level and by studying antibody-secreting cells (ASC) with or without antigen stimulation. Although the manuscript is well-written and presented data are new and interesting, there are several issues that need to be clarified or discussed prior to publication.

Major points:

1.       Introduction should be more concise and clearly present the main goal of the study. The idea of using COBRA HA antigens should be clearly explained.

2.       What was the reason of comparing these two particular immunization routes? How can it be translated to clinical settings?

3.       There was no control group of mice immunized with non-adjuvanted antigens. Can the data on the IgG isotype usage be attributed to the adjuvant rather than to the antigen itself?

4.       Is it relevant to use only one representative strain for H1 and H3 subtypes? The COBRA HAs are artificial immunogens and the HI antibody can be of different magnitude for different H1 and H3 strains. In addition, it would be more relevant to use natural HA antigens for stimulation of mouse splenocytes, as the artificial antigens do not represent a real-life situation when the immunized subject is to be infected with natural flu virus.

Minor points:

1.       Lanes 36-37. FluBlock is a recombinant HA vaccine, not subunit.

2.       Lane 144 – the authors state that the mice were immunized intramuscularly, while later (lane 152) two routes of immunization are described.

3.        How the live splenocytes were obtained from the frozen spleen? It should be described in the Methods section.

4.       Lane 169. ELISA was performed...

Author Response

First of all we would like to thank the editor and the reviewers for their comments which definitely helped in improving the quality of our manuscript.

Please below is reported a point-by-point response to the reviewer's criticisms (following the same numbering).

Major points:

  1. As recommended by the reviewers, the length of the introduction has been reduced and improved. The rationale of using COBRA HAs has been explained in lines 106-110.
  2. We thank the reviewer for having raised this point and giving us the opportunity to clarify it. As explained in the manuscript, these two immunization routes have been chosen since they are the most used in animal pre-clinical studies (especially in mice). There is no intention to translate our methodology and observations to the clinical settings. Our main aim is to provide the investigators working in the field with a guidance for the analysis of the B cell response in a pre-clinical setting (lines 55-64). The rationale of using the IM and IP route has been explained in the introduction (lines 106-110) and in the discussion (lines 387-390).

  3. We thank the reviewer for this comment and having given us the opportunity to clarify this point. Immunization of influenza naïve animals with unadjuvanted antigens would result in a barely detectable antibody response both at the serological and ASC levels. We have clarified this in the introduction (lines 65-69). The observed predominant IgG1 response is mainly associated with the route of immunization (especially the IP route), with the mouse strain (DBA/2J) and the class of the adjuvant utilized. These factors have been reported in lines 375-379.

  4. We thank the reviewer for this comment and for giving us the opportunity to clarify it. The breadth of HAI activity elicited by COBRA HA immunization (Y2 and NG2) has been previously described. In this work we thus limited our HAI assays to the main representative H1N1 and H3N2 strains to evaluate the magnitude of the HAI activity from a functional point of view. We also want to clarify that antigens have not been utilized for the stimulation of the B cells (their stimulation happened in vivo during the final boost) but for detecting the HA-specific antibody response (at the serological and ASC levels). We utilized the COBRA HA Y2 and NG2 as detection Ags since they are the same Ags that were utilized for the immunizations and we thus assessed the homologous response to these Ags at the serological and ASC levels. Using heterologous (or wt Ags) would have resulted in a different level of the antibody response (given the discrepancies between the immunization and detection Ags employed) and thus complicating our interpretation. The rationale of using the COBRA HA Ags has been clarified in the introduction (lines 106-110).

Minor points:

  1. We thank the reviewer for this comment. As per definition recombinant antigens utilized for vaccination are subunit vaccines. However, we have added the word "recombinant" to better clarify the nature of this type of vaccine (line 36-37).
  2. We thank the reviewer for this comment. The discrepancy has been amended accordingly.
  3. We thank the reviewer for this comment. The methodology for obtaining live splenocytes from frozen spleens has been added in the Materials and Methods section (lines 129-134).
  4. We thank the reviewer for having caught this error. The typo has been corrected accordingly.

Reviewer 2 Report

It is an interesting study showing the kinetic of the antibody response following AddaVax-adjuvanted immunization with recombinant influenza antigens. Using the DBA/2J mouse model to  assess the antibody kinetic, the authors showed that a homogeneous kinetic of  the antibody response IgG after IM and IP administration. The authors then assessed the antibody secreting cells from the mice splenocytes using  a FluoroSpot platform, and they found that the  kinetic of the antigen-specific IgG secreting cells peaking at day 5  and day 9 following the IP and IM immunization, respectively.

I have some questions

1- Why the kinetics of ASCs (spleen, fluorospot) is different from kinetic of antibody response in the blood (ELISA). 

2- The authors missed the effect of T cells and cytokines on the antibody production and kinetics.

3-High quality images are required for figure 5, also please add details in legend about the magnification power used

Author Response

First of all we would like to thank the editor and the reviewers for their comments which definitely helped in improving the quality of our manuscript.

Please below is reported a point-by-point response to the reviewer's criticisms (following the same numbering).

  1. We thank the reviewer for having given us the opportunity to clarify this point. We have explained this difference in the introduction and in the discussion (lines 86-90 and lines 399-403).
  2. We thank the reviewer for having brought this point to our attention. As specified in the manuscript we focused on the analysis of the B cell response. Additionally, as explained in the discussion (lines 430-433) we reported this as a limitation of our study. Additionally, as recommended by the academic editor the manuscript type has been changed to a Communication, given the specific focus of our work.
  3. We thank the reviewer for having caught this point. Figure 5 has been substituted with a higher quality image. As communicated by the Immunospot device specialist, there is no magnification involved during the acquisition of the image. The magnification is only digital for visualization purposes. The format of the acquired plate has been added in the figure legend.

Reviewer 3 Report

The authors evaluated the kinetic, magnitude and IgG isotype usage of the antibody response at the serological and at the cell level (antibody-secreting cells) following the intramuscular (IM) and intraperitoneal (IP) administration of a combination of H1N1 and H3N2 next-generation influenza vaccines. Although there are limitations such as lack of memory B cell response longevity and lack of lymph nodes and bone marrow data, the informations obtained from this study would help the people in the field (The COBRA Y2 and NG2 Ag-specific polyclonal antibody responses were mainly composed by IgG1 antibodies. Serum hemagglutination inhibition activity was not affected by the time point of sample collection and the peak of spleen ASCs was affected by the route of immunization). However, the following points must be addressed before publication.

1. The Introduction is too long. It can be reduced to succintly raise interest of the readers and be more clear about the issues. Some of them can be rearranged into Discussion section.

2. Grammatical errors including the following must be corrected.

line 143: ad libitum must be italicized. 

line 199:  for 30 or 1h -> for 30 min or 1 h

Author Response

First of all we would like to thank the editor and the reviewers for their comments which definitely helped in improving the quality of our manuscript.

Please below is reported a point-by-point response to the reviewer's criticisms (following the same numbering).

  1. We thank the reviewer for this comment and having given us the opportunity to improve the manuscript. As recommended, the length of the introduction has been reduced and improved and some sections have been rearranged in the Discussion section.
  2. We thank the reviewer for having caught the grammatical errors. We have amended them accordingly.

Round 2

Reviewer 1 Report

The authors corrected their manuscript according to the reviewers' comments. I only have one note that there is still no information how the splenocytes were revived prior to the cell enumeration for ASC assay. They added information about splenocyte freezing procedure, but not about revival.

Author Response

We thank the reviewer for having given us the opportunity to clarify the splenocyte thawing and revival procedure. This procedure is described in lines 138-142.

Reviewer 2 Report

No further comments

Author Response

We appreciate the reviewer's help in improving the overall quality of our manuscript.